# A Communication-Efficient, Privacy-Preserving Federated Learning Algorithm Based on Two-Stage Gradient Pruning and Differentiated Differential Privacy

**DOI:** 10.3390/s23239305

**Published:** 2023-11-21

**Authors:** Yong Li, Wei Du, Liquan Han, Zhenjian Zhang, Tongtong Liu

**Affiliations:** 1School of Computer Science and Engineering, Changchun University of Technology, Changchun 130012, China; liyong@ccut.edu.cn (Y.L.); 2202103022@stu.ccut.edu.cn (W.D.); 2202103042@stu.ccut.edu.cn (Z.Z.); 2202103086@stu.ccut.edu.cn (T.L.); 2AI Research Institute, Changchun University of Technology, Changchun 130012, China; 3School of Computer Science and Technology, Jilin University, Changchun 130012, China

**Keywords:** differentiated differential privacy, federated learning, gradient pruning, privacy preserving

## Abstract

There are several unsolved problems in federated learning, such as the security concerns and communication costs associated with it. Differential privacy (DP) offers effective privacy protection by introducing noise to parameters based on rigorous privacy definitions. However, excessive noise addition can potentially compromise the accuracy of the model. Another challenge in federated learning is the issue of high communication costs. Training large-scale federated models can be slow and expensive in terms of communication resources. To address this, various model pruning algorithms have been proposed. To address these challenges, this paper introduces a communication-efficient, privacy-preserving FL algorithm based on two-stage gradient pruning and differentiated differential privacy, named IsmDP-FL. The algorithm leverages a two-stage approach, incorporating gradient pruning and differentiated differential privacy. In the first stage, the trained model is subject to gradient pruning, followed by the addition of differential privacy to the important parameters selected after pruning. Non-important parameters are pruned by a certain ratio, and differentiated differential privacy is applied to the remaining parameters in each network layer. In the second stage, gradient pruning is performed during the upload to the server for aggregation, and the final result is returned to the client to complete the federated learning process. Extensive experiments demonstrate that the proposed method ensures a high communication efficiency, maintains the model privacy, and reduces the unnecessary use of the privacy budget.

## 1. Introduction

The Internet of Things (IoT) [1,2,3] is increasingly integrated into our lives, but it also brings about prominent security issues. To address these concerns, federated learning (FL) has been developed. FL enables distributed clients to learn and train data locally while collaborating on shared statistical models. It has gained significant attention in recent years due to its ability to enhance privacy and communication efficiency compared to traditional centralized machine learning approaches [4]. FL ensures data privacy by keeping it local on each client, allowing individual clients to conduct model training using their own data [5,6,7]. The trained model parameters are then uploaded to the server for aggregation, eliminating the need to transmit raw data. However, FL still faces privacy and security challenges, particularly in terms of model parameter privacy. To tackle this issue, differential privacy (DP) has been proposed [8]. DP provides a rigorous privacy guarantee by injecting noise into the parameters to ensure privacy protection. However, excessive noise can lead to a significant reduction in model utility, which can be counterproductive.

The second challenge in FL is the communication cost. FL is now being used in the IoT, which has a particularly large and complex network. The cost of communication between various devices can be very high, making it necessary to find a solution to this problem [9,10,11]. Although Deep Neural Networks (DNNs) have achieved good performances in many real-world scenarios, their high computation and memory requirements make them unsuitable for wider applications on resource-limited devices [12]. Adding noise using DP not only decreases the model accuracy but also increases training costs and time, resulting in significant communication costs. The existing solutions include gradient pruning and model compression. Gradient pruning is relatively easy to implement and effectively removes unnecessary parameters from the network and fine-tunes the model. Currently, mobile and embedded devices are becoming more adapted to DNNs, but each device may have different data, leading to a non-Independent and Identically Distributed (non-IID) data distribution [13]. How can we ensure a high communication efficiency while maintaining the privacy of FL models?

To tackle the aforementioned challenges in FL models, this paper presents a communication-efficient, privacy-preserving FL algorithm based on two-stage gradient pruning and differentiated differential privacy. The proposed algorithm addresses the privacy concern of the pruned model by incorporating DP. It consists of two main components: gradient pruning and ensuring the model privacy through the addition of DP to the pruned model.


**Model pruning**


The IsmDP-FL model pruning technique involves two stages of gradient pruning, the first stage during local model training and the second stage during server aggregation after local training. The experimental results demonstrate that the proposed method outperforms one-stage gradient pruning schemes, such as local model gradient pruning or the server-side gradient pruning of aggregated models.


**Differentiated differential privacy**


The DP algorithm proposed in IsmDP-FL is specifically designed for the pruned small model, ensuring its privacy by incorporating DP after pruning. Following the initial allocation of DP to each network layer of the small model, the remaining parameters are allocated DP again based on their importance. Important parameters receive less noise, while non-important parameters receive more noise, thereby completing the privacy budget allocation of DP. With each round of gradient pruning, the DP allocation for the small model is adjusted, enabling low-memory training while preserving privacy. This approach also reduces the wastage of privacy budget resulting from gradient pruning.


**Contributions**


We introduce a two-stage gradient pruning algorithm, primarily aimed at reducing the communication cost of the model during the federated learning process.We present a differentiated differential privacy algorithm at the network level. Noise is added to the remaining model parameters after each gradient descent, with the amount of noise determined by the importance of these parameters. Initially, differential privacy is allocated to the parameters of each network layer based on their number. Subsequently, differential privacy processing is applied to both important and non-important parameters within each layer, proportionally. Important parameters receive less noise, while slightly more noise is added to other parameters within the same layer.We have conducted multiple experiments on the MNIST, CIFAR-10, and Fashion-MNIST datasets and provided the source code for these experiments.

## 2. Related Work


**Privacy protection for federal learning**


To effectively safeguard the privacy of enterprises and users, FL was proposed as a means to address privacy concerns in machine learning. It is therefore crucial to ensure that training models in FL do not disclose users’ private information.

The issue of data privacy protection was initially proposed by Daren Nusser in the late 1970s. The objective of safeguarding private information in databases is to prevent any user, including both legitimate users and potential attackers, from accessing accurate information about individuals [14]. Subsequently, several privacy protection models with strong practicality were introduced, such as k-anonymity, l-diversity, and t-closeness. However, these models encounter several key challenges [15]: (1) they are vulnerable to background attacks, as demonstrated by incidents like the Netflix privacy breach; (2) integrating them with deep learning poses significant difficulties. In search of a more suitable privacy protection method for FL frameworks, a DP scheme for federated learning was proposed by Yu et al. [16], which effectively combines differential privacy with federated learning. The necessity of safeguarding personal privacy in Natural Language Processing (NLP) [17] systems led to the adoption of DP. DP offers a rigorous privacy definition, and prior work has established DP frameworks without adversarial inference capabilities [18], enabling the limitation of privacy loss for individual data subjects through the addition of noise. Within the FL context, two variants of DP can be employed: (1) Local Differential Privacy (LDP) [19], where each participant adds noise before transmitting updates to the server and (2) Central Differential Privacy (CDP) [20], where the server applies the DP aggregation algorithm.

DP currently faces several challenges [21]. Firstly, challenges in data release include the following: (1) DP for complex and high-capacity network structures; (2) DP on high-dimensional data; (3) DP for correlated data; (4) DP for high-speed data. Secondly, challenges in machine learning include the following: (1) model dependency; (2) the decrease in FL model usability caused by adding noise for DP. For data release, Xiao et al. [22] proposed a multidimensional DP histogram publishing algorithm called DPCube, which supports counting queries of both the unit length and long range in multiple dimensions. This algorithm partitions the original dataset using cell partitioning techniques and adds appropriate Laplace noise to the statistics of each cell. Then, all the cells with added noise are post-processed using a kd-tree structure, i.e., re-partitioning, to obtain a multidimensional V-optimal histogram. To mitigate the impact of privacy protection on model accuracy, Fed-SMP utilizes a new technique called Sparse Model Perturbation (SMP). In this technique, local models are first perturbed by additive Gaussian noise. Additionally, Xu et al. [23] proposed two methods, namely NoiseFirst and StructureFirst, which are used to compute DP-compliant histograms that support long-range count queries and have high query accuracies. In recent works, there have been new attempts and explorations. The combination of DP and homomorphic encryption in [24] provides better privacy protection and a better balance between privacy and efficacy. However, the computational power and cost required for using HE are higher. He et al. [25] proposed an LDP method that takes into account the budget explosion problem that may occur when adding differential privacy. By using adaptive gradient clipping and other methods, the balance between privacy and utility is ensured.

Previous research on privacy protection in FL has primarily concentrated on safeguarding the entire model. While this approach can offer some level of privacy protection for FL models, it often leads to increased training expenses and slower training speeds. Specifically, incorporating uniform noise throughout the entire model using DP not only demands significant training costs but also diminishes the model usability due to the introduced noise.


**Model compression of federated learning**


In large-scale FL, the training cost can be prohibitively expensive, necessitating the exploration of methods to reduce this cost. Adding noise through DP not only diminishes the model usability but also increases training expenses and time. Consequently, the communication cost and efficiency in FL pose additional challenges, prompting various studies to propose solutions. Popular techniques for reducing communication costs in FL include gradient pruning, model compression, and more. Weight pruning, quantization [26], sparsification, and structure pruning are common model compression techniques [27]. For example, ref. [28] reduces communication costs in federated learning by quantizing transferred data. In [29], parameter compensation after training is avoided through the complementary sparsification of parameters between clients and the server. In the field of natural language processing, ref. [30] introduces federated pruning to decrease communication costs. Channel pruning, a form of structural pruning, effectively reduces the computational operations (flops) and memory requirements of the original model without any post-processing steps. Gradient pruning, in contrast, is a relatively straightforward and effective method that involves removing unnecessary parameters from the network and fine-tuning the model. To mitigate communication costs, a novel time-correlated sparsification (TCS) [31] scheme is introduced, leveraging the concept that a sparse communication framework can identify the most crucial elements in the underlying model. Therefore, TCS seeks to establish a correlation between sparse representations used in consecutive iterations in FL, which can significantly reduce the overhead resulting from encoding and transmitting sparse representations without affecting the test accuracy. Additionally, gradient quantization [32] is considered to decrease the bit width of each parameter transmitted. Another solution is to use Top-k sparsification to approximate the gradient matrix using Top-k terms. In recent work, a new FL framework called FedPrune [33] is proposed for efficient communication and personalization. Specifically, under this newly proposed FL framework, each client trains a convergent model locally to obtain key parameters and substructures that guide the network to participate in FL pruning. FedPrune can greatly reduce the communication overhead while achieving a high precision, but it requires each client to learn a personalized model, and the model compression method used may cause model accuracy degradation and model privacy security issues. To effectively address these challenges, the authors propose PruneFL—a novel FL method with adaptive and distributed parameter pruning. This method can adapt to the model size during FL to reduce the communication and computation overhead and minimize the overall training time while maintaining a similar accuracy to the original model. However, PruneFL does not consider the privacy issues associated with the pruned model. In [34], an adaptive dynamic pruning scheme was employed, which retains only the gradients of significant parameters during the training process and achieves promising results. The implementation of gradient pruning not only reduces the model communication costs but also provides certain privacy protection. In [35], a large gradient pruning scheme was designed, ensuring data privacy effectively by employing state-of-the-art attack methods that render the reconstructed images unrecognizable.


**The novelty of our work**


To address the privacy concerns, a network-layer-based DP algorithm is proposed for the DP algorithm. This algorithm is divided into two parts to allocate the privacy budget more precisely and accurately. By incorporating DP into the pruned small model, IsmDP-FL not only reduces training costs but also ensures the security of model privacy.

## 3. Preliminary

In this chapter, we will present the background knowledge relevant to our research. Section 3.1 will provide an introduction to the FL model, offering a comprehensive understanding of the entire FL process. In Section 3.2, we will delve into the concepts and applications of differential privacy, particularly its relevance to FL models. Finally, in Section 3.3, we will discuss the concepts and techniques associated with gradient pruning.

### 3.1. Federated Learning

In a federated learning task, we consider a system with N clients. Initially, the server sends the original model to each participating client. Then, each client n∈N performs local training using their respective local dataset $D_{n}$. During this process, the empirical risk that each client has to minimize is denoted as $F_{n}$.
(1)$$F_{n}\left(w\right)=\frac{1}{D_{n}}\sum_{i\in D_{n}}f_{i}\left(w\right)$$

In the federated learning system, the training loss function for each local client is denoted as $f_{i}\left(w\right)$. This function measures the discrepancy between the model’s output and the desired output for a given set of input samples $x_{1},\ldots ,x_{N}$. The overall loss function for the entire system is defined as $f\left(w\right)=\frac{1}{N}\sum_{n}f\left(w,x_{n}\right)$, where N represents the total number of clients. During local training, each client optimizes the model parameters using gradient descent. This iterative optimization method adjusts the parameters in the direction of steepest descent of the loss function to minimize the training loss and improve the model’s performance.
(2)$$w_{n}^{t}=w_{n}^{t-1}-\eta \nabla F_{n}\left(w_{n}\right)$$
where $w_{n}^{t}$ represents the parameters of the nth client in the current round and $g_{n}=\nabla F_{n}\left(w_{n}\right)$ represents the gradient of the current client parameters. Then, the client transmits the trained model to the server and performs model aggregation in the server:(3)$$W^{t}=\frac{1}{\left|D\right|}\sum_{n=1}^{N}\left|D_{n}\right|w_{n}^{t}$$
where |D| is the total dataset, $D_{n}$ is the dataset owned by each client, and $W^{t}$ is the global model parameter of the current round of server aggregation. For complex networks, the loss function f(w) is usually non-convex and difficult to minimize. Minimization is usually achieved using a small-batch stochastic gradient descent algorithm. A batch of random samples is formed at each step and calculated as $g_{B}=\frac{1}{\left|B\right|}\sum_{x\in B}\nabla_{w}f\left(w,x\right)$ to estimate the gradient $\nabla_{w}f\left(w\right)$. The objective of a federated learning system is to find an optimal set of parameters that minimize the global empirical loss. The aggregated model, which reflects the collective knowledge of all participating clients, is then sent back to each client.

### 3.2. Differential Privacy

Differential privacy was initially proposed by Dwork in 2008. It employs a rigorous mathematical framework and the randomized response technique to guarantee that the dataset remains below a certain threshold when the output information is influenced by a single record. This ensures that no third party can make accurate inferences based on the output, making it one of the most secure methods among disturbance-based privacy protection approaches. In our specific context, the formal definition of differential privacy is as follows:

**Definition** **1.***If, for any two adjacent inputs $x,x^{\prime }\in X$ and any output subset S⊂R, a random query M on a training set with domain X and range R has the same output results on two adjacent inputs and satisfies the equation below [36,37], then the random mechanism M is said to satisfy (ε,δ)–differential privacy:*(4)$$P_{r}\left(M\left(x\right)\in S\right)\leq e^{\varepsilon }P_{r}\left(M\left(x^{\prime }\in S\right)+\delta \right)$$where ε represents the privacy budget; when ε = 0 and δ = 0, mechanism M possesses perfect privacy. In the context of differential privacy, when the input x∈X is perturbed or changed, the query function M may reveal sensitive information about the local data, making it challenging to maintain ε–differential privacy. To analyze the mechanism’s privacy, we use the concept of sensitivity of the query function. Sensitivity is defined as follows:

**Definition** **2.**
*d is a positive integer, and D is a collection of datasets. $f:D\to R^{N}$ is a function. The Δf representative $l_{1}$-sensitivity is as follows:*

(5)
$$\Delta f=max||f\left(x_{1}\right)-f\left(x_{2}\right){\left|\right|}_{1}$$



**Definition** **3.**
*For $f:D\to R^{N}$, the $l_{2}$-sensitivity of f is defined as follows:*

(6)
$$\Delta_{2}f=max||f\left(x_{1}\right)-f\left(x_{2}\right){\left|\right|}_{2}$$



Subsequently, we introduced the addition of noise to the parameters in the federated learning model. We employed two mechanisms, namely the Laplacian mechanism and the Gaussian mechanism. Initially, we defined the Laplacian mechanism as follows:

**Definition** **4.**
*Given a dataset D, there is a function f, whose $l_{1}$-sensitivity is Δf (Definition 2). Then, the random algorithm M=f(D)+L provides (ε,0)–differential privacy protection, where $L∼Lap(0,\frac{\Delta f}{\varepsilon })$ is the added random noise probability density function, obeying μ=0, $\lambda =\frac{\Delta f}{\varepsilon }$.*


The Laplacian mechanism is a commonly used privacy protection mechanism for numerical queries due to its simplicity in implementation. It provides (ε,0)–differential privacy by adding noise following the Laplacian distribution $L(0,\frac{\Delta f}{\varepsilon })$ to the query results. Now, we define the Gaussian mechanism:

**Definition** **5.**
*Given a dataset D, there is a function f, whose $l_{2}$-sensitivity is $\Delta_{2}f$ (Definition 3), for any δ∈(0,1), $\sigma =\sqrt{2Ln\frac{1.25}{\delta }}\times \frac{\Delta_{2}f}{\varepsilon }$. Then, the random algorithm M=f(D)+L provides (ε,δ)-differential privacy protection, where $L∼Gaussian(0,\sigma^{2})$ is the probability density function of the added random noise, which obeys the parameter μ=0, $\sigma =\sqrt{2Ln\frac{1.25}{\delta }}\times \frac{\Delta_{2}f}{\varepsilon }$.*


Both the Gaussian mechanism and the Laplacian mechanism are oriented to numerical query results, and noise can be added directly to the results.

### 3.3. Gradient Pruning

There are numerous techniques available for gradient pruning, such as structured pruning, unstructured pruning, iterative pruning, and adaptive gradient pruning. In FL models, pruning can significantly reduce training costs and enhance the training speed. However, it may also lead to some accuracy loss in the model. Therefore, it is essential to identify an optimal pruning scheme that can effectively reduce training costs while minimizing the impact on the model accuracy. Pruning entails the removal of redundant channels, neuron nodes, and network layers from the network, resulting in a lighter network without compromising the performance, as illustrated in Figure 1.

The following are some different classifications of pruning:Based on the type of network components, pruning can be categorized into neuron pruning and connection pruning;Based on whether the network structure undergoes changes before and after pruning, it can be classified into structured pruning and unstructured pruning;Based on whether pruning is conducted during the inference stage, it can be categorized into static pruning and dynamic pruning.

## 4. Method

In Section 4.1, we present the complete workflow of our method. In Section 4.2, we explain the fundamental concept behind the implementation of the pruning algorithm. In Section 4.3, we outline the core idea of the differential privacy algorithm. Lastly, in Section 4.4, we provide a detailed description of our proposed algorithm and present the corresponding pseudocode.

### 4.1. Workflow of IsmDP-FL

The participants involved in the federated learning process include clients and servers. Our proposed method consists of two main stages:On the client side, the first stage involves initializing the model pruning before training. This includes removing redundant elements from the model. After pruning, we add differential privacy noise to the remaining parameters. The noise-added parameters are then transmitted to the server for further processing;The second stage involves pruning the model further based on the received parameters and transferring the remaining parameters back to the client.

The workflow of IsmDP-FL is shown in Figure 2.

This scheme utilizes the fundamental framework of federated learning. The entire workflow involves the following aspects:(1)The server sends the initial global model.(2)The first-stage model pruning is performed on the initial model sent by the server, and the pruned model is passed to the client.(3)Each client trains a local model based on its local data.(4)Noise is added to the model of each client, and the noise-added models are uploaded to the server. The server then aggregates these models.(5)After model aggregation on the server, the aggregated model undergoes a second-stage gradient pruning. The pruned models are then passed back to the respective clients.(6)Steps 3–5 are repeated until the model converges.

We will provide detailed explanations of the specific implementation schemes for model pruning and differential privacy in the following chapters.

### 4.2. Model Pruning Process in IsmDP-FL

Our approach starts with a common federated learning process. First, each client n receives parameters from the server. Before training the model locally, we first prune the client model parameters through the model pruning function Prune($w_{t}$) to obtain the pruned model parameter $w_{t}^{\prime }$. After local training, it is uploaded to the server for aggregation, and then we also use the model pruning function Prune($W_{t}$) to perform model pruning on the aggregated parameters, and we can obtain a new aggregation model parameter $W_{t}^{\prime }$:(7)$$W_{t}=\sum_{n=1}^{N}\frac{|D_{n}|}{\left|D\right|}w_{t}^{\prime }$$
(8)$$w_{t}^{\prime }=Prune\left(W_{t},q\right)$$
where q is the compression ratio of the model compression.

### 4.3. Differentiated Differential Privacy Implementation in IsmDP-FL

The method we propose involves applying differential privacy processing to each layer of parameters in the model. During the federated learning process, after pruning the model, the parameters $w_{t}^{\prime }$ of the local client are processed on a per-layer basis. The importance measure of each layer parameter is determined by its absolute value. We calculate the median, denoted as *h*, of the absolute values of all parameters in each layer. If the absolute value of a parameter is less than *h*, we add a large amount of noise to it. Conversely, if the absolute value is greater than or equal to *h*, we add a smaller amount of noise. Our specific approach involves sorting the noise to be added based on the absolute value and creating a new noise sequence. We then add noise of a corresponding magnitude to the model parameters based on their importance.

### 4.4. Algorithm Description

The pseudocode for our proposed IsmDP-FL is presented in Algorithm 1. The initial global model of the server is trained on the server side, based on the set model sparse ratio. Local training is then performed on the client side, using local data (lines 5–8). The model parameters trained by each client are subjected to differential privacy noise for privacy protection (lines 16–22), and parameter aggregation is performed on the server side (lines 9–10). Finally, model pruning is carried out on the aggregated parameters (line 11), and the resulting model is transferred back to the client until the federated learning model converges. On the client side, the dataset is first divided into batches (line 24). Then, the gradients of each batch of data are computed, followed by gradient clipping (line 26). Subsequently, gradient descent training is performed, and noise is added to the parameters during training for privacy protection (lines 27–30).

Algorithm 2 outlines our proposed model pruning algorithm. First, we sort the parameters of each layer based on their absolute values and obtain the sorted parameter sequence. Then, we determine the pruning threshold R by taking the absolute value of the parameter at the position obtained by the percentage q of this sequence (lines 3–5). We prune the parameters with absolute values smaller than the threshold R, which means we keep only the parameters with absolute values greater than or equal to the threshold R. Next, we generate a mask matrix consisting of only 0s and 1s (lines 6–14) according to the selected pruning threshold. Finally, we obtain the final sparse model by performing element-wise multiplication between the current parameters and the complement of the mask matrix (line 15).
**Algorithm 1** IsmDP-FL1:**Procedure ServerExecute:**2:require server model sparsity q3:**Initialize:** t = 0, $W_{0}$ randomly, tensor mask to zero4://server-side main program5:$w_{0}^{\prime }\leftarrow Prune\left(W_{0},q\right)$6:**if**
 T==0 
**then**7:    **for** eachclientn **do**//In Parallel8:        $\left(\theta_{n},D_{n}\right)=n.ClientUpdate\left(w_{0}^{\prime }\right)$9:    **end for**10:**end if**11:**if**
 T!=0 
**then**12:    **for** t∈T **do**13:        **for** eachclientn **do**//In Parallel14:           $\left(\theta_{n},D_{n}\right)=n.ClientUpdate\left(w_{t}^{\prime }\right)$15:        **end for**16:         $\left|D\right|=\sum_{n}\left|D_{n}\right|$17:         $W_{t+1}\leftarrow \sum_{n=1}^{N}\frac{|D_{n}|}{\left|D\right|}\theta_{n}$18:         $W_{t+1}^{\prime }\leftarrow Prune\left(W_{t+1},q\right)$19:    **end for**20:**end if**21:**Procedure ClientUpdate(*w*)**22://Executed at Clients23:require learning rate η, privacy budget ε, gradient norm bound C24:$D_{n}\leftarrow local\;data\;divided\;into\;minibatches$25:**for**
 $each\;batch\;B\in D_{n}$ 
**do**26:    **Compute gradient:** $g_{B}=\nabla F_{n}\left(w;B\right)$27:    **Clip gradient:** $g_{B}^{\prime }\leftarrow g_{B}/max\left(1,\frac{\left|\right|g_{B}{\left|\right|}_{2}}{C}\right)$28:**end for**29:$\theta_{n}\leftarrow w-\eta g_{B}^{\prime }$30:$\theta_{n}^{\prime }\leftarrow Addnoise\left(\theta_{n}\right)$31:**return**
 $\left(\theta_{n}^{\prime },|D_{n}|\right)$

**Algorithm 2** Prune(w, q)
1:require tensor mask to zero2://runs on client and server3:

$w_{xin}\leftarrow sort\left(|w|,descending=False\right)$

4:**for**
 $each\;layer\;in\;w_{xin}$ 
**do**5:    $R\leftarrow q^{th}\;percentile\;in\;w_{xin}$6:
**end for**
7:**for**
 eachelemente∈w 
**do**8:    **if** |e|<R **then** e←09:    **end if**10:
**end for**
11:

mask←w

12:**for**
 eachelemente∈mask 
**do**13:    **if** e!=0 **then** e←114:    **end if**15:
**end for**
16:

w←w⨀¬mask

17:**return**
 w


Algorithm 3 outlines our implementation of adding noise to the parameters. Firstly, we sort the randomly generated noise matrix to obtain two noise matrices: one sorted from large to small and one sorted from small to large (lines 3–4). Next, we add noise to each parameter based on the selected threshold processing approach. Specifically, we add elements from the small noise matrix to important parameters and elements from the large noise matrix to non-important parameters (lines 6–11). Finally, we obtain the parameters after adding noise. We allocate noise based on the noise matrix generated for each layer of parameters. First, we sort the noise matrix based on the absolute values of the noise parameters. The large noise matrix has absolute values arranged in descending order, while the small noise matrix has absolute values arranged in ascending order. According to the threshold value h that we determine, we add parameters with absolute values smaller than the threshold h using the large noise matrix starting from the beginning. Parameters with absolute values greater than or equal to the threshold h are added using the small noise matrix starting from the beginning.
**Algorithm 3** Addnoise(w)**Input:** Examples $x_{1},\ldots ,x_{n}$, random noise matrix noise. Parameters: h, Pick a threshold for adding large or small noise.1:**Initialize:** $\theta_{0}$ randomly2:**for**
 t∈[T] 
**do**3:    $Noise_{small}\leftarrow sort\left(|\mathrm{noise}|,descending=False\right)$4:    $Noise_{big}\leftarrow sort\left(|\mathrm{noise}|,descending=True\right)$5:**end for**6:**for**
 $each\;element\;e\in \theta_{t}$ 
**do**7:    **if** e<hande!=0 **then** Add the $Noise_{small}$ matrix to e from the first8:    **end if**9:    **if** e≥hande!=0 **then** Add the $Noise_{big}$ matrix to e from the first10:  **end if**11:**end for**12:**Output:** Parameter $\theta_{T}$ after adding noise

## 5. Experiment

### 5.1. Experiment Settings

**Datasets**. We utilized the following datasets for our experiments:

(1) MNIST dataset: It comprises a total of 70,000 images, with 60,000 images allocated for training and 10,000 images for testing. Each image is a grayscale image with a size of 28 × 28 pixels, depicting a handwritten digit ranging from 0 to 9.

(2) CIFAR-10 dataset: This dataset contains 50,000 training images and 10,000 testing images. The images are RGB color images with a size of 32 × 32 pixels. The dataset consists of images belonging to ten different classes, namely airplanes, automobiles, birds, cats, deers, dogs, frogs, horses, ships, and trucks.

(3) Fashion-MNIST dataset: It consists of 60,000 training images and 10,000 testing images. The images depict ten different fashion items, including T-shirts/tops, trousers, pullovers, dresses, coats, sandals, shirts, sneakers, bags, and ankle boots. Similar to the MNIST dataset, each image in the Fashion-MNIST dataset is a grayscale image with dimensions of 28 × 28 pixels.

The MNIST and Fashion-MNIST datasets are grayscale images, which are single-channel grayscale images. CIFAR-10, however, is an RGB three-channel color image dataset.

**Models**. For the MNIST dataset, we opted for a CNN network architecture comprising two convolutional layers and one fully connected layer. The dimensions of the fully connected layer are 32*7*7. For the Fashion-MNIST dataset, we employed the VGG-16 network, which consists of 13 convolutional layers and 3 fully connected layers. The purpose of using this model is to increase the network’s depth, which can have an impact on the final performance of the network. As for the CIFAR-10 dataset, we utilized a CNN network architecture with 3 convolutional layers and a fully connected layer. Additionally, we incorporated a dropout layer into the network to mitigate overfitting during the training process.

**Parameter setting**. For the MNIST dataset, we configured the FL environment with 10 participants. We set the local training epochs to 10 and the communication rounds between the clients and the server (epoch) to 200. We used the SGD optimizer with a learning rate (lr) of 0.01, which adapts during the model training process. For the CIFAR-10 dataset, we also configured the FL environment with 10 participants. We set the local training epochs to 10 and the communication rounds between the clients and the server (epoch) to 500. We used the SGD optimizer with a learning rate (lr) of 0.001, which also adapts during the model training process. For the Fashion-MNIST dataset, we again set the FL environment with 10 participants. We set the local training epochs to 10 and the communication rounds between the clients and the server (epoch) to 200. We used the SGD optimizer with a learning rate (lr) of 0.0001, which adapts during the model training process. Regarding differential privacy protection, we separately set the privacy budget to 1 for each client to apply differential privacy to the model.

**Evaluation metrics**. The primary evaluation metrics comprise the loss (the loss function’s value during training), accuracy (the final test results), and θ (the number of communication rounds required to achieve the target accuracy).

We have divided the experiment into two parts: one part focuses on the model accuracy with differential privacy, while the other part examines the model accuracy using the model compression method. Therefore, our experiment is divided into two sections, each using the following comparison algorithms:Differential privacy: in this part, we employ the classic differential privacy algorithm DP-FL and the local differential privacy (LDP) algorithm.Model compression: for this part, we choose the recently proposed quantization method JoPEQ and the complementary sparse (CS) method.

Due to the different settings of the baseline algorithm itself during the experiment, the JoPEQ algorithm does not involve non-IID dataset configuration and a random selection of clients. Therefore, all settings remain consistent in our experiments. To present our experimental results more clearly, we have presented the results of experiments at different sparsity levels in tables. In the figures, we showcase the best accuracy achieved by each method.

### 5.2. Experimental Results of Our Proposed Method

#### 5.2.1. Comparison of Model Compression Schemes

Here are the results of the FedAvg algorithm on three datasets: 98.76% (MNIST), 86.55% (Fashion-MNIST), and 58.95% (CIFAR-10). We compare our method with the JoPEQ scheme utilizing quantization and the scheme employing complementary sparsification. Our proposed method exhibits a lower training loss and a slightly improved model accuracy. The experimental results on the MNIST dataset are presented in Figure 3. Figure 3a illustrates the comparison of the model accuracy between our method and the baseline, while Figure 3b provides a zoomed-in view of the comparison of the method and baseline model accuracies.

In Figure 3, we observe that our method’s model convergence is similar to that of the FedAvg method. Compared to other baselines, our method’s model convergence is slightly better. In Figure 3b, we can see the final model accuracy, where our method achieves the best performance of 99.01%. When the lattice size is 1 in the JoPEQ method, the accuracy is slightly higher than FedAvg. However, the JoPEQ method’s model training process is relatively unstable. Although the CS method is stable, its final accuracy is not as good as our proposed method.

In Table 1, we present a comparison of the model accuracies achieved by the CS, JoPEQ algorithm, and our method at different sparsity levels. Among the four methods on these three datasets, IsmDP-FL consistently achieves the highest accuracy when the model sparsity ranges from 0.1 to 0.9. It is not that IsmDP-FL performs the best at every sparsity level, but rather it achieves the optimal accuracy within the range of 0.1 to 0.9 (the best accuracy among all schemes on the same dataset).

In Table 1, we can observe that our method outperforms CS. An interesting observation is that for the CIFAR-10 dataset, our highest accuracy is achieved when 90% of the parameters are removed. Our IsmDP-FL method shows a slight improvement over the FedAvg algorithm, and with increasing pruning rates, the accuracy sometimes even improves instead of decreasing. We attribute this to the complexity of the dataset, where for non-IID data, the model accuracy may exhibit some uncertainty.

We have also presented the experimental results on the Fashion-MNIST dataset in Figure 4 and the CIFAR-10 dataset in Figure 5.

In Figure 4, we can observe that our method achieves a final accuracy of 86.40%, which is only 0.15% lower than the FedAvg algorithm. However, the two quantization methods of JoPEQ exhibit a low accuracy in the early stages of the curve. Our analysis suggests that this may be due to some information being affected by the quantization process. Nevertheless, the convergence of JoPEQ is better than the other methods. In Figure 4a, we observe that the accuracy of the JoPEQ method does not improve until after 30 rounds of training. This may be due to the impact of quantizing the data in the early stages of training. After 30 rounds of training, the impact on the model becomes smaller and smaller, and the curve eventually converges. The experiments on the CIFAR-10 dataset in Figure 5 show that our method outperforms the baseline scheme by approximately 1% compared to the FedAvg algorithm. However, the JoPEQ quantization method does not perform well in this case.

#### 5.2.2. Comparison of IsmDP-FL and Existing Differential Privacy Schemes

The methods compared in this study include the local differential privacy scheme used in JoPEQ and the classic DP-FL differential privacy scheme. We set the privacy budget to 1 and apply Laplacian noise and Gaussian noise to the model, respectively. Experiments are conducted on the MNIST dataset. The comparison results of model accuracy are presented in Figure 6. Figure 6a illustrates the accuracy comparison with Laplacian noise, while Figure 6b demonstrates the experiment with the addition of Gaussian noise.

In Figure 6a, we observe that under the same privacy budget (Laplacian noise), with 200 rounds of communication, our method achieves the highest final model accuracy, albeit only 0.1% higher than the DP-FL method. The LDP method performs the worst in this setting. Although the final model converges, the curve of the model exhibits significant fluctuations during the training process. Moving on to Figure 6b, we can see that in the case of Gaussian noise, the LDP curve converges slowly. However, the final accuracy is higher than that of the DP-FL and IsmDP-FL methods. Both the DP-FL and IsmDP-FL methods demonstrate better model convergence effects. Additionally, even after removing 40% of the parameters, the accuracy of the IsmDP-FL method remains above 96%.

We present the experimental results on the Fashion-MNIST and CIFAR-10 datasets in Figure 7 and Figure 8, respectively.

In Figure 7 and Figure 8, we compare the performance of adding Laplacian noise and Gaussian noise to the CIFAR-10 dataset and the Fashion-MNIST dataset, respectively. In Figure 6, it is evident that on the CIFAR-10 dataset our method outperforms the baseline method DP-FL with Laplacian noise by 8.25%. Moreover, we have achieved this improvement while removing 80% of the parameters, resulting in a significant reduction in the communication cost. However, the accuracy of JoPEQ’s LDP method is greatly reduced. Additionally, we conducted experiments with different sparse ratios and found that the accuracy does not decrease gradually but starts to decline after reaching the highest point. We attribute this phenomenon to the complexity and uncertainty of the dataset. Moving to Figure 8, we conducted experiments on the Fashion-MNIST dataset. In the case of Laplacian noise, our method achieves a slightly lower accuracy compared to the LDP method. However, we were able to remove 40% of the parameters, resulting in a reduced communication cost. In contrast, when Gaussian noise is applied, our method performs equally as well as the LDP method, but we are able to remove 70% of the parameters, leading to a significant reduction in the model size. In Table 2a, we present the model accuracy of IsmDP-FL after adding differential privacy under different sparsity ratios. The table demonstrates that even after removing a large number of parameters we still achieve a high accuracy. Furthermore, in Table 2b, we provide a comparison of the best accuracy achieved by three differential privacy algorithms under different datasets and noise types while maintaining a fixed privacy budget of 1.

Based on the results, we can observe that in the MNIST dataset, IsmDP-FL outperforms DP-FL and LDP under Laplace noise. However, LDP demonstrates the best performance under Gaussian noise. In the Fashion-MNIST dataset, LDP consistently performs the best, while IsmDP-FL shows relatively good results. In the CIFAR-10 dataset, IsmDP-FL significantly outperforms DP-FL and LDP methods.

#### 5.2.3. Security Analysis of Differential Privacy Algorithms

In this section, we evaluated the privacy protection capabilities of differential privacy methods across three datasets. We primarily employed a well-known gradient inversion attack method (DLG attack) to attack federated learning models trained on these datasets. We trained the federated learning models for 10, 50, and 100 rounds, respectively, and then observed the level of blurring in the attacked original images.

Below, we present the results of attacking the trained models of FedAvg, DP-FL, IsmDP-FL, and LDP methods on the MNIST, Fashion-MNIST, and CIFAR-10 datasets. The differential privacy noise applied in our experiments is Laplace noise. First, we focus on the experimental results for the MNIST dataset, as shown in Figure 9. Our results demonstrate the attack effectiveness for training rounds of 10, 50, and 100. Figure 9a illustrates the attack effect on the FedAvg method’s trained model using DLG. Figure 9b displays the attack effect on the trained model of the DP-FL method. Figure 9c showcases the attack effect on the trained model of the LDP method. Finally, Figure 9d presents the attack effect on the trained model of the IsmDP-FL method.

Next, we will present the results for the Fashion-MNIST and CIFAR-10 datasets, as shown in Figure 10 and Figure 11.

Based on the above results, we can first observe that when DLG attacks MNIST images, the three differential privacy methods have almost equal protection capabilities for the images. When Fashion-MNIST images are attacked, the privacy protection capability of the DP-FL method is weaker than the LDP and IsmDP-FL methods. For CIFAR-10 images, we can see that the protection capability of the LDP method is significantly weaker, while the privacy protection capabilities of the IsmDP-FL and DP-FL methods are stronger. In other words, our method is even better than DP-FL. This also confirms the results obtained in our accuracy comparison above.

## 6. Conclusions

This paper mainly introduces a communication-efficient, privacy-preserving FL Algorithm based on two-stage gradient pruning and differentiated differential privacy, named IsmDP-FL. Firstly, we propose a novel differential privacy algorithm to address the privacy concerns in federated learning. Secondly, we focus on the federated learning model and tackle the issue of communication costs by employing a two-stage adaptive gradient pruning approach. This method prunes the gradients both during local training and during aggregation on the server. We have achieved promising results on the MNIST, CIFAR-10, and Fashion-MNIST datasets. Compared to the baseline, our approach has a slightly improved model accuracy while reducing the communication cost.

In terms of future work, we aim to further optimize differential privacy in our model. One direction is to explore the application of client-level differential privacy, which can enhance privacy protection and better accommodate model compression and local differential privacy techniques. This will enable us to minimize communication costs in the federated learning process while ensuring privacy. Additionally, we plan to investigate the level of privacy protection provided by our method. We intend to evaluate its resilience against existing classic attack methods and demonstrate the effectiveness of our model’s defense mechanisms. By conducting such analyses, we can gain insights into the robustness and security of our approach.

## Figures and Tables

**Figure 1 sensors-23-09305-f001:**
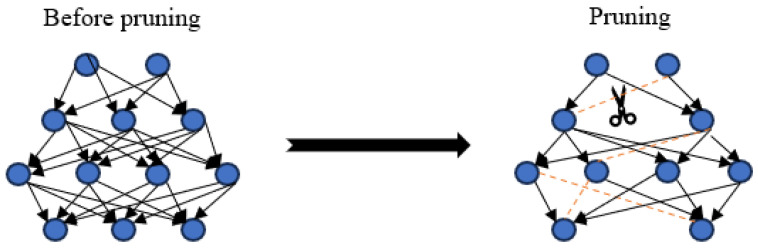
Schematic diagram of neural network pruning.

**Figure 2 sensors-23-09305-f002:**
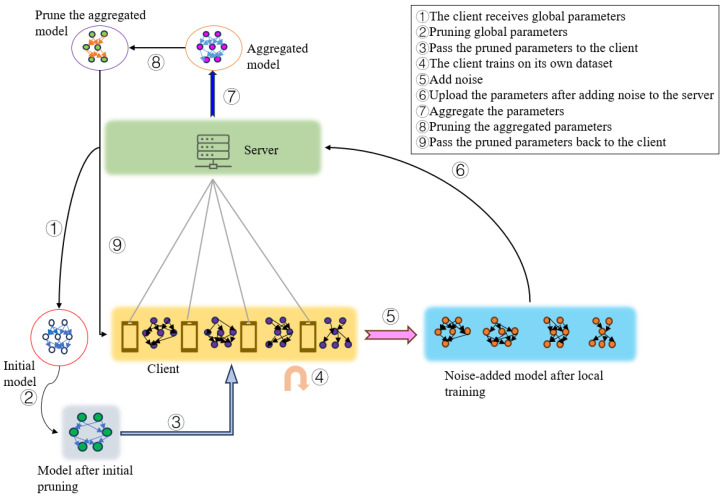
Workflow of our proposed method.

**Figure 3 sensors-23-09305-f003:**
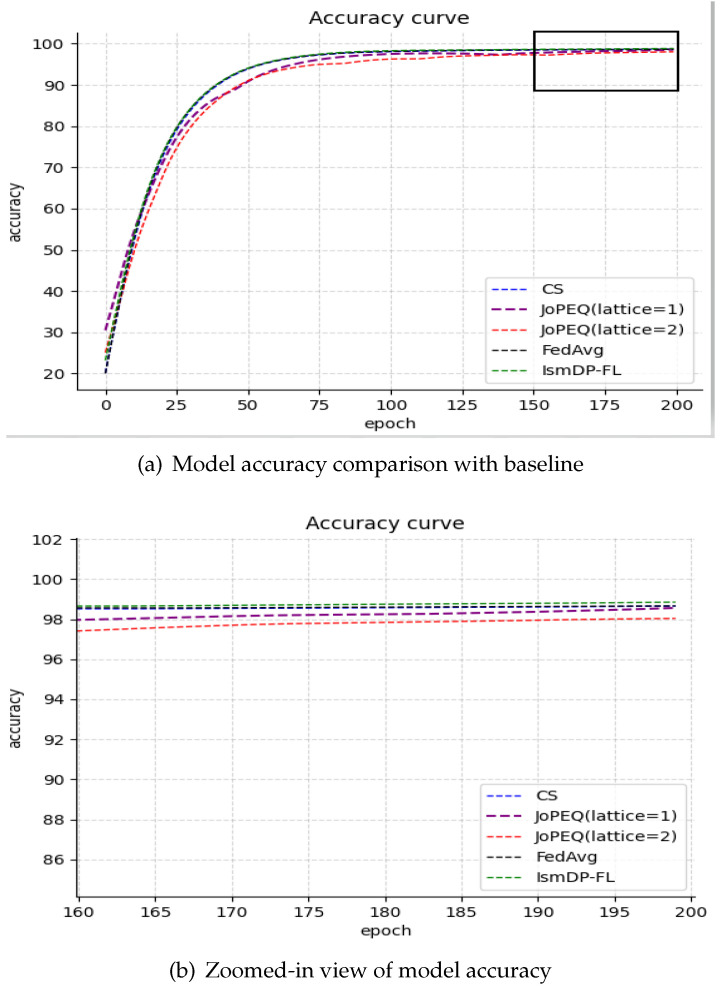
Our method is compared with existing pruning algorithms. In the JoPEQ method, two quantization methods are set, which are divided into lattice sizes of 1 and 2; the other is a complementary sparse method, which shows the accuracy of the FedAvg method.

**Figure 4 sensors-23-09305-f004:**
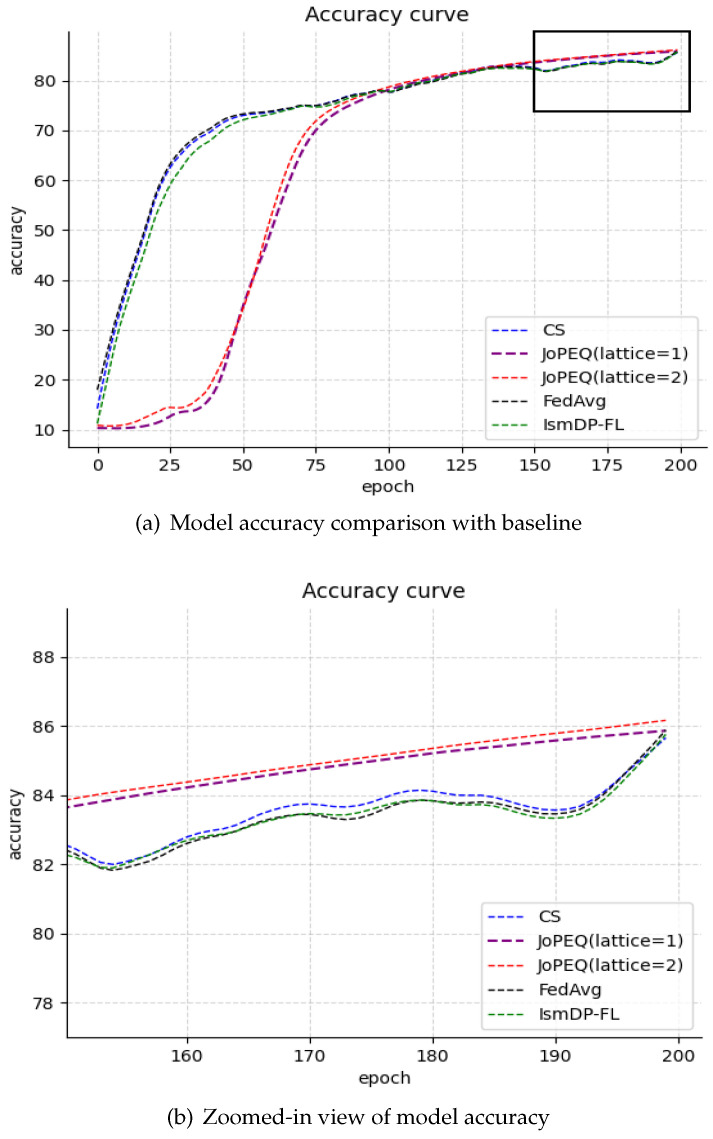
Our experiments on the Fashion-MNIST dataset.

**Figure 5 sensors-23-09305-f005:**
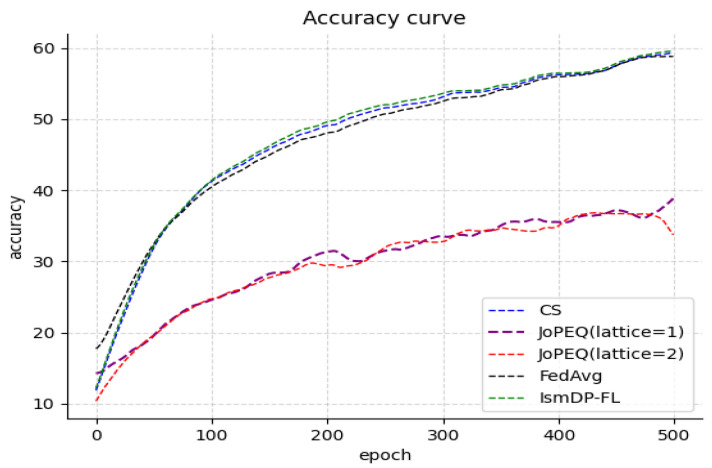
Our experiments on the CIFAR-10 dataset. Our method outperforms the baseline.

**Figure 6 sensors-23-09305-f006:**
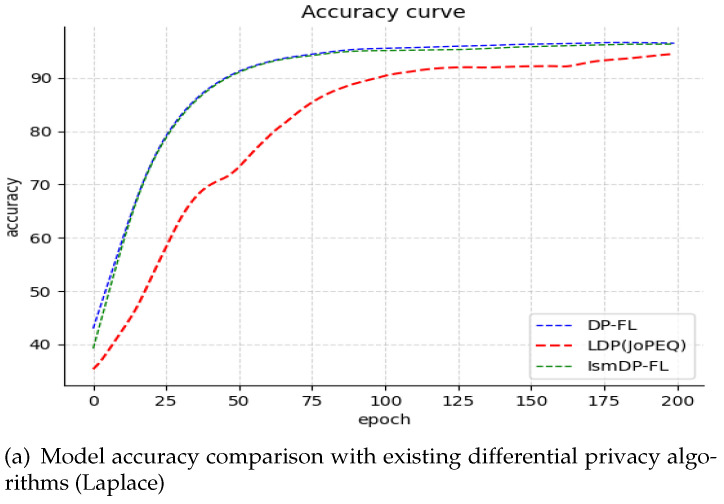
Our method is compared with the DP-FL algorithm as well as the LDP algorithm in JoPEQ. We set the privacy budget size to 1 and the learning rate to 0.01, with Laplacian noise and Gaussian noise.

**Figure 7 sensors-23-09305-f007:**
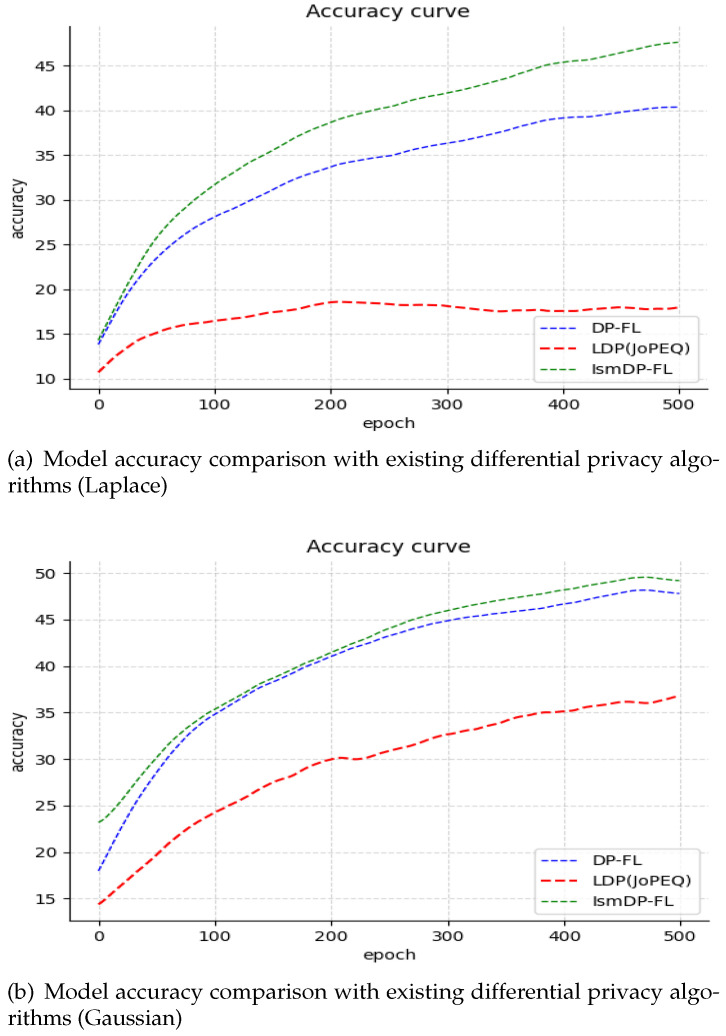
Our experiments on the CIFAR-10 dataset. Our method outperforms the baseline.

**Figure 8 sensors-23-09305-f008:**
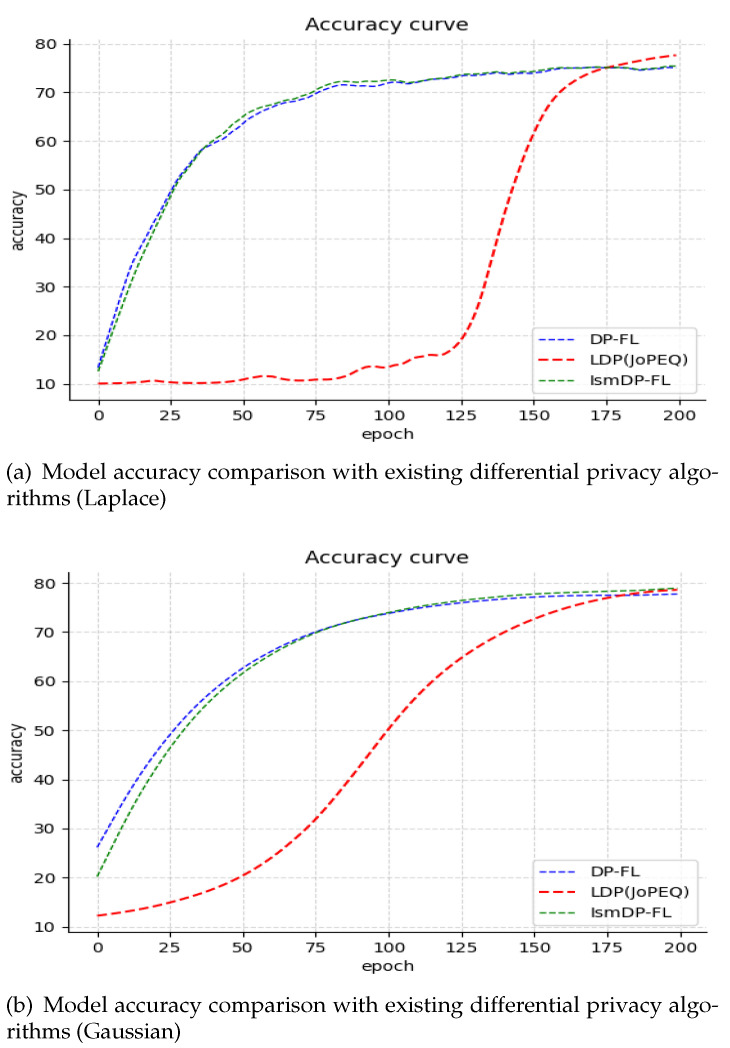
Our experiments on the Fashion-MNIST dataset.

**Figure 9 sensors-23-09305-f009:**
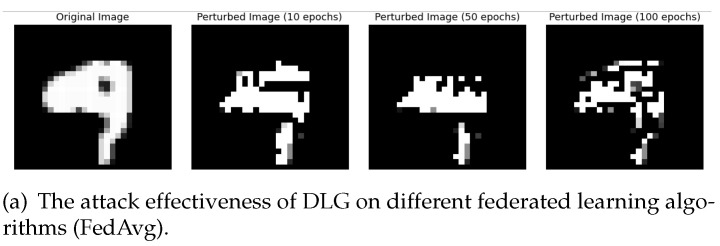
The effect of DLG on attacking the FedAvg model as well as the trained models with three differential privacy algorithms.

**Figure 10 sensors-23-09305-f010:**
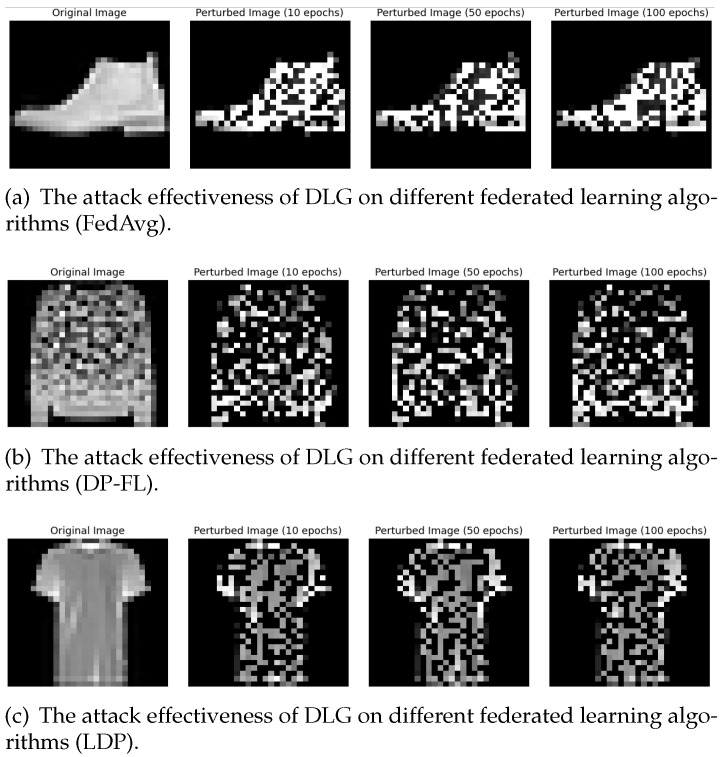
The effect of DLG on attacking the FedAvg model and the trained models with three differential privacy algorithms on the Fashion-MNIST dataset.

**Figure 11 sensors-23-09305-f011:**
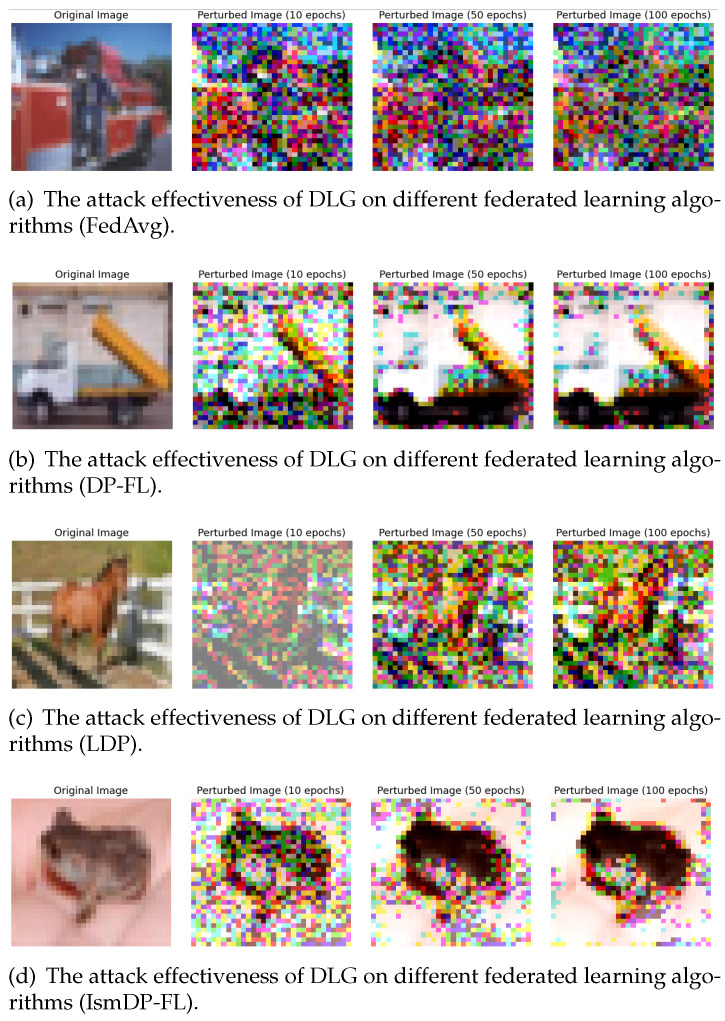
The effect of DLG on attacking the FedAvg model and the trained models with three differential privacy algorithms on the CIFAR-10 dataset.

**Table 1 sensors-23-09305-t001:** Accuracy of models with different sparsities in different datasets.

Sparsity		0.1	0.2	0.3	0.4	0.5	0.6	0.7	0.8	0.9
MNIST	CS	98.76	98.59	98.66	98.72	98.48	98.26	97.69	97.34	93.50
	IsmDP-FL	**99.01**	98.65	98.65	98.69	98.16	98.01	97.93	97.44	93.73
	JoPEQ (Lattice = 1)	98.83	98.88	98.93	98.96	98.94	98.96	98.96	98.97	-
	JoPEQ (Lattice = 2)	98.23	97.98	98.09	97.99	98.03	98.05	98.10	98.07	-
Fashion-MNIST	CS	86.3	85.51	85.91	86.08	86.12	86.07	85.76	84.02	83.82
	IsmDP-FL	86.34	85.52	85.69	**86.40**	86.27	86.37	85.57	83.37	83.12
	JoPEQ (Lattice = 1)	85.71	85.96	85.73	85.75	85.67	85.81	85.73	85.74	-
	JoPEQ (Lattice = 2)	86.28	86.29	86.02	86.02	85.96	86.01	85.96	86.00	-
CIFAR-10	CS	58.79	58.29	57.08	58.47	58.7	58.77	59.97	59.45	58.81
	IsmDP-FL	59.31	58.38	58.81	59.43	59.18	59.45	58.00	60.09	**60.11**
	JoPEQ (Lattice = 1)	41.71	40.61	40.26	40.94	40.84	40.70	40.96	40.97	-
	JoPEQ (Lattice = 2)	27.61	28.73	28.18	28.26	28.31	28.67	28.32	28.32	-

**Table 2 sensors-23-09305-t002:** Comparison between our method and LDP (JoPEQ) and DP-FL algorithms with a fixed privacy budget of 1. In (a), we give the model accuracy of our method under different model sparsity ratios. In (b), we compared the best model accuracies of the three algorithms.

(a) The accuracy of IsmDP-FL model under different model sparsity ratios										
**Sparsity**		**0.1**	**0.2**	**0.3**	**0.4**	**0.5**	**0.6**	**0.7**	**0.8**	**0.9**
MNIST	Laplace	95.90	95.88	95.88	96.01	**96.37**	95.04	95.08	95.87	92.09
	Gaussian	96.01	93.90	95.69	**96.18**	95.30	94.16	94.72	94.83	91.77
Fashion-MNIST	Laplace	74.68	75.69	74.76	**75.73**	74.40	73.95	74.13	73.76	73.95
	Gaussian	80.23	80.25	80.90	82.26	80.76	79.26	**82.47**	81.86	81.22
CIFAR-10	Laplace	40.31	39.86	40.50	41.86	42.52	44.38	46.95	**48.72**	47.27
	Gaussian	46.33	46.52	47.83	**48.00**	46.38	46.33	46.25	47.34	45.34
(b) Comparison of the best accuracies of three differential privacy algorithms when the privacy budget is fixed to 1										
Method		DP-FL			LDP			IsmDP-FL		
MNIST	Laplace	96.27			95.56			**96.37**		
	Gaussian	95.19			**97.35**			96.44		
Fashion-MNIST	Laplace	75.23			**77.9**			75.73		
	Gaussian	79.93			82.47			**82.47**		
CIFAR-10	Laplace	40.47			19.26			**48.72**		
	Gaussian	46.56			40.01			**48.00**		

## Data Availability

Data are contained within the article.

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
