# Peer review of "A Communication-Efficient, Privacy-Preserving Federated Learning Algorithm Based on Two-Stage Gradient Pruning and Differentiated Differential Privacy"

_sensors, 2023, doi:10.3390/s23239305_

Round 1

Reviewer 1 Report

Comments and Suggestions for Authors

The purpose of this paper is to address privacy issues and communication overhead issues in federated learning. For the privacy-preserving problem, this paper designed a Differentiated Differential Privacy algorithm, which adds less noise to important parameters, and slightly more noise is added to other parameters within each layer. For the communication-efficient problem, this paper proposed a two-stage gradient pruning algorithm to reduce the communication cost. However, there are some problems needed to be solved.

- For the two-stage gradient pruning algorithm, there is no explanation for how to get the model sparsity q, nor does it explain what model parameters are pruned. In addition, the server prunes the initialization model parameters and then prunes the aggregate model parameters for each round, which is far-fetched to call these two steps two-stage pruning, the two operations do not indicate a significant difference.

- Teoretically, the pruning operation will affect the accuracy of the model, but the experiments in this paper show that the two-stage pruning operation in this paper has no effect on the accuracy of the model, which should be theoretically proved.

- For algorithm 2, there is no text description, the readability is poor, and lines 7 and 12 contradict with each other.

- For the Differentiated Differential Privacy algorithm, “The absolute value of the parameter is used as the importance measure of the parameter, and then the median h of the absolute value of each layer parameter is taken as the threshold value. If the absolute value of the parameter is less than h, the parameter is added to large noise; if the absolute value of the parameter is greater than or equal to h, the parameter is added to little noise.” How are large noise and small noise defined?

- What is the purpose of including differential privacy in this article? Added differential privacy to prevent inference attacks on gradient parameters by which entity? There is no security analysis for whether it would cause privacy leakage after adding noise.

- Please pay attention to word spelling problems, such as the title of this article “Communication-efficient” is misspelled.

- Some of the step subjects in the workflow section of the article are unclear and poorly readable.

Comments on the Quality of English Language

The editorial quality of this paper should be significantly improved.

Author Response

Thank you for your advice. Please see the attached document for my response.

Reviewer 2 Report

Comments and Suggestions for Authors

The introduction of the paper needs improvement to motivate the proposed work better. For example, sentences such as  "Therefore, many studies have proposed methods such as Gradient Pruning and Model Compression to solve this issue." are introduced out of context. 

The related works section should be more focused. Currently, it provides a generic view of the area (especially on Privacy protection for federal learning; there is much less discussion on the privacy protection of FL). 

It's not clear what the authors mean by "When ε= 0, the above definition will be satisfied ε–differential privacy.". The authors need to understand that When ε=0 and δ=0, the mechanism M will have perfect privacy. Besides, the phrasing "and represents the probability of breaking this limit" can be misleading. δ is the probability of model failure. Besides, the phrases such as "Laplace of distributed", do not make sense. 

Further discussion on the algorithm is necessary, as the authors have only provided an abstract view of it. Specifically, the pruning and noise addition parts require a more detailed explanation. It is unclear how the proposed pruning and noise addition method enforces differential privacy. The authors should provide formal proof of the differential privacy guarantees.

The figure quality should be improved. 

The authors are encouraged to conduct a more detailed analysis of the privacy budget consumption and model performance under different privacy settings.

Comments on the Quality of English Language

The grammar of writing needs to be improved. 

Author Response

(The authors gave the same response as above.)

Reviewer 3 Report

Comments and Suggestions for Authors

Acyually, the main concept of Differential privacy in federated learning was presented by following paper that authors should discuss on it and add in the reference list:

Yu, S., Cui, L. (2023). Differential Privacy in Federated Learning. In: Security and Privacy in Federated Learning. Digital Privacy and Security. Springer, Singapore. https://doi.org/10.1007/978-981-19-8692-5_5

Second, the related work section should be extended by addressing some ne published papers to show advantages and weaknesses of recent case studies.

Third, for Definition 3.1, authors should add relevant references for this explanation.

Also, in Algorithm 1, Authors should explain Procedure ClientUpdate(w) exactly.

For Section 5, Authors applied both grayscale image and RGB color images for their experiments. So, what is different between these types ? Also, authors can discuss on them that their DP-FL method is useful for which type of images.

Authors should exactly present evalaution results of accuracy for all three datasets.

Comments on the Quality of English Language

Should be imporved.

Author Response

(The authors gave the same response as above.)

Round 2

Reviewer 3 Report

Comments and Suggestions for Authors

Authors have successfully addressed all comments in this revision.